# Corrosion Behavior of TiMoNbX (X = Ta, Cr, Zr) Refractory High Entropy Alloy Coating Prepared by Laser Cladding Based on TC4 Titanium Alloy

**DOI:** 10.3390/ma16103860

**Published:** 2023-05-20

**Authors:** Liang Liu, Hongxi Liu, Xiaowei Zhang, Yueyi Wang, Xuanhong Hao

**Affiliations:** School of Materials Science and Engineering, Kunming University of Science and Technology, Kunming 650093, China; zhxw72@163.com (X.Z.); wyy_0710@126.com (Y.W.); hxh_instinct@163.com (X.H.)

**Keywords:** refractory high entropy, laser cladding technology, microstructure, corrosion resistance

## Abstract

TiMoNbX (X = Cr, Ta, Zr) RHEA coatings were fabricated on TC4 titanium alloy substrate using laser cladding technology. The microstructure and corrosion resistance of the RHEA were studied by XRD, SEM and an electrochemical workstation. The results show that the TiMoNb series RHEA coating was composed of a columnar dendrite (BCC) phase, a rod-like second phase, a needle-like structure and equiaxed dendrite, but the TiMoNbZr RHEA coating showed high-density defects, similar to those in TC4 titanium alloy, which were composed of small non-equiaxed dendrites and lamellar α’(Ti). In the 3.5% NaCl solution, compared with TC4 titanium alloy, the RHEA had a lower corrosion sensitivity and fewer corrosion sites, showing better corrosion resistance. The corrosion resistance of the RHEA ranged from strong to weak in this order: TiMoNbCr, TiMoNbZr, TiMoNbTa and TC4. The reason is that the electronegativity of different elements is different, and the speeds of the formation of the passivation film were very different. In addition, the positions of pores appearing in the laser cladding process also affected the corrosion resistance.

## 1. Insroduction

Ti-6Al-4V (TC4) titanium alloy, with low density, high specific strength, strong weldability and other excellent properties, is widely used as an important raw material in aerospace and other high-tech fields [1,2]. However, the lack of corrosion resistance of TC4 titanium alloy in extreme environments means it struggles to meet the requirements of aerospace development in terms of material properties, which greatly limits its application in practical engineering applications [3,4]. To improve the performance of TC4 titanium alloy, surface modification technology is the most popular method to form coatings with excellent performance on the surface of TC4 titanium alloy [5,6,7,8]. Due to the clear distinction between solvent and solute, the types of elements added in traditional alloys are limited and their content is small, meaning the performance of the material has struggled to meet the requirements of aviation work. However, RHEAs generally have principal elements n ≥ 5, and are allotted according to the equal molar ratio or approximate equal molar ratio. The characteristics of each principal element and the interaction between atoms give the alloy a complex “cocktail” effect, so it shows better performance, such as in corrosion resistance [9,10,11,12,13]. Compared with other surface modification technologies, laser cladding technology has the advantages of fast cooling speed, small heat-affected zone, a wide selection of cladding materials, and economic and environmental protection, so it is more popular [14,15,16,17,18]. By laser cladding a RHEA coating on the surface of titanium alloy, the advantages of the high strength and plasticity of the titanium alloy can be combined with the corrosion resistance of RHEA, which not only solves the defects of titanium alloy, but also greatly reduces the production cost and consumption in service.

At present, there is little research on the corrosion resistance of high-entropy alloys composed of refractory metal elements, but their excellent performance cannot be ignored. Jiang [19] et al. prepared TiZrAlNb and TiZrAlNbCo high-entropy alloy coatings on the surface of TC4 titanium alloy using laser cladding technology. Through a study of their microstructure and corrosion resistance, it was found that the TiZrAlNb and TiZrAlNbCo high-entropy alloy coatings were, respectively, BCC and BCC-fcc matrix phases, and were accompanied by a small number of intermetallic compounds; the dendritic crystals of the TiZrAlNbCo alloy coating were much finer than those of the TiZrAlNb alloy coating. Compared with the TC4 substrate and the TiZrAlNb alloy coating, the TiZrAlNbCo alloy coating has excellent corrosion resistance (Icorr is 3.66 × 10^−9^ A/cm^2^). The fundamental reason for microstructure optimization and performance improvement is the appearance of the FCC phase in the TiZrAlNbCo alloy coating. Huang [20] et al. synthesized a new refractory high-entropy alloy coating (RHEC) with a composition close to that of TiNbZrMo on 316L via laser cladding. The corrosion resistance of TiNbZrMo RHEC was tested and the results show that RHEC is a single-phase solid solution with a body-centered cubic (BCC) structure, and it exhibits a typical dendritic microstructure. Electrochemical studies show that the corrosion resistance of RHEC in a 3.5 wt. % NaCl solution is significantly better than that of 316L stainless steel. Li [21] prepared NbMoTaTi with a single BCC solid solution structure on a molybdenum plate using laser cladding technology, with an average hardness of up to 397 HV and excellent mechanical properties at high temperatures. Although the passivation films formed by Ta, Cr and Zr have good corrosion resistance [22,23,24,25], no studies have compared the corrosion resistance of different elements Ta, Cr and Zr. We added Ta, Cr and Zr elements to a TiMoNb alloy to prepare the RHEA coating. The TiMoNb alloy phase has a BCC single-phase structure, which is conducive to the formation of RHEAs with a simple phase structure. By studying the corrosion resistance of the TiMoNbX (X = Ta, Cr, Zr) four-element RHEA after adding different refractory elements, we hope to deepen the understanding of the corrosion resistance mechanism of RHEA, and provide new theoretical data on and ways to improve the corrosion resistance of TC4 titanium alloy.

## 2. Experimental Methods

### 2.1. Materials Preparation

The TC4 titanium alloy sheet was cut into a cuboid small matrix with the size of 40 mm × 13 mm × 4 mm using a wire cutting machine. The substrate was sand, produced with a grinding machine with 200# and 400# sandpaper to remove the oxide layers on the surface. The roughness of the surface of the substrate after sanding was Ra 6.3~50, which was beneficial to reducing the laser reflection of the substrate in the preparation process. The substrate surface was washed with anhydrous acetone, Keller’s reagent (HF:HCI:HNO_3_:H_2_O = 2:3:5:9), clean water and anhydrous ethanol in order to remove the surface oil. The washed substrate was cleaned in an ultrasonic cleaning machine containing anhydrous ethanol, and the surface of the substrate was dried using a hair dryer and then put into a drying oven for use. TiMoNbX (X = Cr, Ta, Zr) RHEA was prepared using single-pass laser cladding technology. The purity of each original powder used for sample preparation was greater than 99.95%. Various powders were weighed and mixed at equal molar ratios, and placed in the MSK-SSFM-13S high-flux planetary ball mill for vacuum ball-grinding. The ratio of pellet to material was 3:1; the grinding ball was a stainless steel ball with a diameter of 5 mm, and the grinding time was 1 h. The alloy powder was obtained after full mixing, and the particle size of the powder was 150–300 mesh. At a mass fraction of 96%, the alloy powder was mixed with 4% anhydrous ethanol, and stirred until reaching the sticky state. Using the laminate method, the powder was pressed on the surface of the TC4 matrix with a special mold to form a preset layer. The thickness of the prefabricated layer was 1.0 mm, and it was placed in a drying oven at 60 °C for constant temperature treatment for 6 h. The anhydrous ethanol was volatilized, and then the cladding layer could be obtained using laser cladding. The parameters of the laser cladding process were as follows: laser power 2000 W, scanning speed 600 mm/min, spot diameter 2 mm, defocusing capacity 50 mm, argon gas, gas flow 8 L/min. The surface phase composition of the cladding coating samples was analyzed using a D/max-3BX (Rigaku Corporation of Japan) X-ray diffractometer (XRD) (Cu as cathode target, Ka wavelength 1.54186A). The tube pressure of the XRD ray diffraction analyzer was 40 kV, the tube flow was 15 mA, the diffraction angle range was 20~90°, the scanning rate was 10 (°)/min, the scanning step was 0.02°, the scanning interval was 1 s/step, and the continuous scanning mode was used. A Zeiss-EVO18 (Carl Zeiss AG) scanning electron microscope and its accompanying BRUKER-Xflash (Bruker GMBH in Germany) energy spectrometer were used to conduct SEM tests on the corroded samples. The purpose was to analyze the microstructure and element distribution of the coating by magnifying different areas of the coating, followed by point scanning and surface scanning. Before observation, the samples were treated with 240~2000# SiC abrasive paper and polished with 1.0 μm diamond abrasive paste. The sample was then washed in alcohol for 10 min with an ultrasonic cleaner. Finally, it was etched for 25 s with an etching solution (HF:HNO_3_:H_2_O = 1:2:7).

### 2.2. Electrochemical Corrosion Test

A VSP multi-channel electrochemical workstation was used to measure the polarization curve. The sample to be tested was the working electrode, using the Pt sheet electrode as the opposite electrode, and the reference electrode was the saturated calomel electrode. Before the experiment, copper wire was connected to the sample to be tested using an electric soldering iron, and this was then coated with epoxy resin, such that the surface of the sample to be tested was exposed. After 200 mesh, 400 mesh and 600 mesh sandpaper treatments, 800 mesh sandpaper was used to water-grind the sample’s surface until there were no obvious scratches. Then, the sample was examined under a LeicaDFC280 (Leica of Germany) metallographic microscope, and finally, mechanical polishing was performed with a polishing cloth. The sample was ultrasonically cleaned for later use. The corrosion solution was 3.5% NaCl, the potential scanning range was from −1.2 V to 2.0 V, and the scanning speed was 1 mV/s. For comparison, the matrix TC4 was treated in the same way.

## 3. Results and Discussion

### 3.1. Microstructure and Phase Constitution of TiMoNbX (X = Ta, Cr, Zr) RHEA

Figure 1 shows the XRD pattern of the TiMoNbX (X = Cr, Ta, Zr) RHEA drawn using the Origin2019b software from the measured data. For convenience, the three TiMoNbCr, TiMoNbTa and TiMoNbZr RHEAs are respectively denoted as +Cr, +Ta and +Zr.

It can be observed from Figure 1 that there is one strong peak and four weak peaks in similar positions in the XRD patterns of the RHEA coatings with +Cr, +Ta and +Zr. The strong peaks appeared at 39.2°, while the weak peaks appeared at 36.8°, 40.5°, 56.1° and 70.9°, respectively. It should be noted that there is an additional weak peak at 84.4° in the XRD pattern of the +Ta coating, while the weak peak appears at 77.0° in the XRD pattern of the +Zr coating. The results show that the phase compositions of all coatings are similar and relatively simple, and the three four-element RHEAs are all biphasic, while the main phase is BCC.

By comparing the d values in the XRD pattern with those in the JCPDS card, we can confirm that the main component of all the coatings is β(Ti) with a structure of BCC, accompanied by a small amount of α’(Ti) with a structure of HCP. It is worth noting that Mo, V, Ta, Nb and Cr are all β-phase-stabilizing elements. Although Cr is a β-eutectoid element, no corresponding diffraction peak has been found in the XRD pattern. This is due to the rapid laser cladding process, which causes the cooling rate to be so fast that there is not enough time to form intermetallic compounds. In addition, due to the rapid heating and cooling rate of the coating, α’(Ti) becomes stable below 882 °C, β (Ti) becomes stable above 882 °C, and the transformation between the two occurs at 882 °C. Besides extending the α phase region to a higher temperature, the α + β two-phase region is formed. This causes part of the β (Ti) to turn into α’(Ti) martensite as well, which is why a small amount of α’(Ti) is still present in the coating. It is noteworthy that the diffraction peaks of β(Ti) and TiO_2_ in the +Zr coating coincide at 77.0°.

The peak positions of the +Ta and +Cr coatings did not shift significantly, because the difference in atomic radius between Ta (1.43 Å) and Cr (1.25 Å) was small, so the lattice distortion of RHEA was very small. In contrast, the main peak of the +Zr coating moved to the left, which is due to the large atomic radius of Zr (1.62 Å), which leads to an enhancement of the lattice distortion effect in the coating system, such that the diffraction peak moves to the left. To verify this result, the exact locations of diffraction peaks and corresponding crystal plane indices of the BCC phases of the three coatings were substituted into Brag’s Law (a=λh^2+g^2+l^22sin⁡θ), and the lattice constants a of the three RHEA coatings could be obtained. Table 1 shows the data calculated based on this equation. The lattice index of the +Zr coating is smaller than that of the +Ta and +Cr coatings, so the main peak moves to the left. If the diffraction peak at 77.0° of the +Zr coating is β (Ti), this will lead to a larger lattice index, and the main peak should shift to the right, which is inconsistent with the result in Figure 1, so the diffraction peak at 77.0° should correspond to that of TiO_2_.

There are only two phases in the coating, and this is because for the four-element alloy system, the mixing entropy is greater than the entropy of intermetallic compounds, and so the electronegative difference of the system becomes smaller and the generation of intermetallic brittle compounds is inhibited, such that the RHEA coating forms simple BCC and HCP structures [15,26,27,28].

Figure 2 shows a backscattering picture of the macroscopic morphology of the coating of the TiMoNbX (X = Cr, Ta, Zr) RHEA under SEM. The cross sections of the +Cr, +Ta and +Zr RHEA coatings can be divided into four zones from the top of the coating to the substrate, namely, the cladding zone, the bonding zone, the heat-affected zone and the substrate. As can be seen from Figure 2, there is no crack in the RHEA coating and only a small number of pores. At the same time, the metallurgical bonding effect between the coating and substrate is good, and there are only a few pores on the joint of the coating and the substrate.

The main reason for the existence of pores in the coating is mainly related to the operation of the laser cladding preparation technology, in addition to the unavoidable factors of experimental operation. Specifically, when the cladding powder is wet, oxidized, contains impurities, or there are gaps between the powders, laser beam irradiation will produce gas. Because the pool condenses so quickly, the gas that cannot escape is retained in the coating, forming pores. At the same time, when the flow of laser cladding protective gas is too great, the protective gas will be become part of the molten pool. Due to the small volume of the molten pool and the fast cooling speed, the gas cannot be discharged and will stay in the molten pool, and porosity will be formed.

The XRD test results of the prefabricated powder show that no impurity elements or oxidation were found. After drying, the prefabricated powder did not produce gas during the cladding. Therefore, it is speculated that the laser cladding powder may be oxidized after prefabrication, or there is a void created by the porosity. By observing Figure 2a–c, it can be found that the +Ta and +Cr coatings have few pores in the middle and upper parts of the cladding area, while the +Zr coating has more pores in the middle and upper parts of the cladding zone, which is speculated to be caused by the oxidation of laser cladding powder after prefabrication, or the existence of pores. Moreover, the Mo element can easily form volatile oxides at temperatures higher than 800 °C, which is also one of the reasons for the pores’ development. These pores will affect the corrosion resistance of the coating due to their different locations.

Figure 3 shows the SEM image of the microstructure of the +Cr RHEA coating. Figure 3f shows that the coating is of good quality and metallurgically bonded to the substrate. Figure 3b shows the microstructure of the upper part of the cladding area. The amplification in Figure 3c shows that the top area of the coating is mainly composed of coarse columnar dendrite and the rod-like second phase precipitated from the gray intergranular structure. The secondary dendrite arm of the columnar dendrite is approximately perpendicular to the primary dendrite arm. Columnar dendrites are mainly formed at the top of the coating, and a few form in the middle and lower regions of the coating; the closer to the middle region of the coating, the less frequently they appear. The structural morphology of the middle part of the coating is shown in Figure 3c, which reveals an acicular structure, also known as a basket structure. The growth pattern of the acicular crystals in the middle part of the coating is intricate. The direction of grain growth is affected by many factors, among which the temperature gradient is one of the most important. In the process of crystal growth, the temperature gradient will lead to changes in the distribution of atoms in the crystal, which will affect the direction of crystal growth. In general, crystals grow in the direction of increasing temperature. This is because, in the temperature gradient field, the diffusion rate of substances in the high-temperature region is faster than that in the low-temperature region. Therefore, the nucleation and growth of crystals often occur in the high-temperature region, resulting in a crystal growth direction that is consistent with the temperature gradient direction. At the same time, the high-temperature region is also conducive to the adsorption and diffusion of species on the crystal’s surface, thus contributing to the nucleation and growth of the crystal’s surface. However, in the middle region of the coating, heat dissipation loses its direction, so the generated tissues are not directional. Figure 3e shows the microstructure at the junction of the cladding zone and binding zone. Figure 3a shows that there are mainly a small number of gray columnar dendrites and basket structures, and a small amount of lamellar eutectic structures. Due to the heating and air-cooling of the high-energy laser beam during laser cladding, the β (Ti) of the thin layer is transformed, and the relative content of other elements also increases. Although the RHEA is beneficial to forming a simple solid solution structure due to the high mixing entropy, the complex interaction of various principal elements means that the alloy is composed of various phases, and the eutectic structure is a typical one.

Figure 4 shows the microstructure and EDS plane scanning, expanding on that shown in Figure 3b. The main components of the dendrite are titanium and vanadium. The RHEA itself contains a certain amount of titanium. Further, the titanium element in the TC4 substrate shows good solid solubility with the refractory alloy. Therefore, under the action of a high-energy laser beam, a large amount of titanium in the substrate is diluted and soluble in the coating, and is enriched in the dendrite region, while the vanadium element is dispersed in the coating. Thus, preferential crystallization gives rise to the BCC phase. Combined with the XRD analysis of the distribution characteristics of elements in the cladding layer, it can be inferred that the coarse dendrites are β (Ti) with a BCC crystal structure. Since the atomic radius of Ti is larger than that of other elements (Mo, V, Al), it will cause lattice instability when assimilated into the BCC phase, resulting in large lattice distortion and an increase in the lattice constant of BCC. The rod-like structure in Figure 3c was analyzed by point scanning, as shown in Table 2. The element present at the highest quantity was Ti. Mo and Cr were also more abundant than other elements. Based on the observation of the distribution and structures of elements, it can be concluded that the BCC phase had a dendritic texture at the junction of the top of the coating and the matrix due to the cooling rate and element segregation, while in the middle of the coating, the BCC phase showed a full β transition structure composed of acicular α and β matrix [29]. A small amount of α’(Ti) was present as a rod-like second phase, and was precipitated in the intergranular structure at the base joint and at the top of the coating.

Figure 5 shows the microstructure of the +Ta RHEA coating. Figure 5f shows the morphology of the joint of the coating and matrix. It can be seen that the coating is of good quality and is in a metallurgical combination with the matrix. Figure 5a shows the microstructure of the upper part of the cladding area, and Figure 5b shows the enlarged local morphology of the microstructure at the top of the coating. In this part of the coating, many columnar dendrites appeared in dark gray, and the bright white, rod-like second phase is the precipitation in the light gray intercrystalline structure. Combined with the EDS element plane scanning shown in Figure 6, we can infer that Ti is the main element in the gray-black dendrite, and the remaining five elements are evenly distributed and less abundant. The results show that a considerable amount of matrix is melted and participates in the formation of coating during the cladding process, and the reason why the content of the Al element is higher than that of other elements is the diffusion of matrix elements at high temperatures, as well as the fact that a small amount of Al element is vaporized. However, due to the rapid cooling, some of the Al gas fails to escape, resulting in the formation of pores. Combined with the XRD shown in Figure 1, we see that the dendrite has the structure of β (Ti) (BCC). Figure 5d shows the morphology of the middle part of the coating. It can be seen from the figure that the structure is basically a basket structure. The point scanning results of the second rod-like phase are shown in Table 3, in which the contents of elements in the two tissues are roughly the same. Except for a large amount of dispersed Al element, the main components of point 1 are Ti, Nb and Ta, with Ti atoms accounting for 74.3%, Nb atoms 5.8% and Ta atoms 5.2%. The main components of point 2 are Ti, Mo and V. Ti atoms account for 77.3%, Mo atoms account for 3.6% and V atoms account for 3.6%. A small amount of the α’(Ti) has no obvious organizational characteristics, and forms HCP with other elements. We can conclude that the change in tissue shape is not only related to the temperature gradient and solidification rate (G/R), but also to the contents of Mo, Nb, Ta, Al and V elements. It is found that the BCC phase presents a columnar dendrite structure at the top of the coating, while the BCC phase present a reticular structure at the junction between the coating and the substrate.

Figure 7 shows the microstructure of the +Zr coating. As can be seen from Figure 7a, the coating is of good quality and metallurgically bonded with the substrate. However, compared with the morphology of +Cr and +Ta, the +Zr coating has obvious porosity defects. In the lower right corner of the coating, the white area represents high-density defects similar to those in the TC4 titanium alloy. The defects are composed of white non-equiaxed dendrites. Figure 7c shows the microstructure of the upper half of the coating area, which is mostly composed of black equiaxial dendrites and the rod-like second phase, and is basically similar to the microstructure of +Cr and +Ta. Combined with the point scanning results shown in Table 4, we can see that the main element of the dendrites is Ti. Combined with XRD results, the coarse dendrite can be identified as β (Ti). Compared with Figure 7b,c, the dendrites in Figure 7b are finer and more concentrated at the same magnification. Lamellar structures appear at the edge of the dendrites, which should be due to the transformation of β (Ti) in the fine lamellae. In order analyze element composition, we performed face scan analysis on this kind of structure. As shown in Figure 8, except for Al and Zr, which contain very little dendritic content, other elements such as Ti, Mo, Nb, Zr and V all show a dispersed distribution. In the middle and lower regions of the coating, the microstructure basically has the morphological characteristics shown in Figure 7e. At the joint of the coating and the matrix, the columnar dendrites grow perpendicularly to the binding interface; this is the beginning of the laser cladding process, and much of the processing heat cannot be released quickly from the coating. The best heat dissipation direction is vertical, and the best grain growth direction is regular. Many of the nucleation elements get in the way of each other, which explains this structure. The BCC phase at the top of the coating contains equiaxial dendrites, while the BCC phase mainly comprises columnar dendrites at the junction between the coating and the substrate, and high-density defects are formed in some areas of the coating.

### 3.2. Electrochemical Analysis of RHEA and TC4

Figure 9a shows the typical dynamic potential polarization curves of TC4 titanium alloy substrate and RHEA TiMoNbX (X = Cr, Ta, Zr), with three refractory elements added at room temperature in the 3.5 mass% NaCl solution. The samples of +Cr, +Ta and +Zr RHEAs show similar electrochemical corrosion responses, which can be divided into four typical stages, namely, the active dissolution stage, the primary passivation stage, the primary breakdown stage, and the secondary passivation stage, while TC4 presents the secondary passivation breakdown stage. At the beginning of the experiment, the active dissolution stage was relatively smooth, and with the increase in potential, the curve became steeper. Then, a primary passivation zone was formed, followed by a secondary passivation zone as the point continued to rise, followed by a breakdown. It is not difficult to see from Figure 9a that, no matter what kind of RHEA is used, compared with TC4, the span of the passivation zone is larger than that of TC4. The passivation zone forms due to the formation of a protective film on the alloy’s surface, thus delaying the further corrosion of Cl^−^. To further evaluate its electrochemical corrosion resistance, the relevant parameters, including Ecorr, Icorr, primary breakdown potential (Epp1) and secondary breakdown potential (Epp2), were studied. It can be seen from Table 5 that the self-corrosion potentials of RHEA coatings with +Cr, +Ta and +Zr are −0.230 V, −0.248 V and −0.277 V, respectively, which are all higher than that of TC4 (−0.397 V), indicating that compared with the TC4 titanium alloy, TiMoNbX (X = Cr, Ta, Zr) RHEAs have lower corrosion sensitivity, or fewer corrosion sites [22].

Figure 9b shows the EIS of the TC4 matrix, +Cr, +Ta and +Zr RHEA coating samples in a 3.5 mass% NaCl solution under OCP conditions. The Nyquist curve of +Ta shows a semicircular segment at frequencies higher than 0.263 Hz, followed by a straight line in the low frequency region, indicating that the dissolution kinetics of passivated films on the sample surface are limited by diffusion in the oxidation products, whereas +Cr, +Zr and TC4 all show semicircular segments. The curvatures of +Cr, +Ta and +Zr are greater than that of TC4, indicating that the coatings of RHEA +Cr, +Ta and +Zr are more resistant to corrosion than TC4 in the 3.5 mass% NaCl solution.

Figure 10 shows the microstructures of three RHEA coatings and the TC4 matrix after corrosion. The three RHEA coatings all show very similar corrosion morphologies after the corrosion test. The surfaces of the +Cr, +Ta and +Zr coatings are basically flat and smooth, and white spots are formed in local areas. Such morphologies are also present in large quantities on the matrix corrosion morphology of TC4, and polygonal particles appear on the matrix. We can infer that this corrosion morphology is formed by the peeling of the passivation film. To further determine the morphology and organizational elements of these white particles and white areas, we enlarged the specific white area and then carried out point scanning, as shown in Figure 11. The elemental content is shown in Table 6. After amplification, the white region on the +Cr RHEA has been shown to present a white petal-shaped equiaxed crystal, and the content of Cr has been relatively increased. The white granular areas on the surfaces of the +Ta and +Zr coatings are composed of a variety of elements, presumably formed by the peeling of a small amount of passivation film on the coating’s surface. The polygonal particles of the TC4 matrix are residual NaCl crystals, as revealed by elemental analysis, and these particles are more likely to be present in the corrosion pit. As regards the morphology after corrosion, there are basically no corrosion pits on the surface of the +Cr coating, the passivation film falls off of the surfaces of the +Ta and +Zr coatings, the white particles formed on the +Zr coating are larger, and more corrosion pits appear on the surface of the TC4. Therefore, corrosion resistance can be ranked, from strong to weak, as +Cr, +Zr, +Ta and TC4.

From the above analysis, we can infer that, compared with the TC4 matrix, RHEA coatings have better corrosion resistance, and the +Cr coating has the best corrosion resistance, mainly because Cr can form a Cr_2_O_3_ passivation film during corrosion, which can effectively prevent the continuous occurrence of corrosion and slow down the corrosion rate. The TaO and ZrO_2_ passivated films were formed from +Ta and +Zr coatings during the corrosion process, and Cr_2_O_3_ showed better corrosion resistance. Due to the presence of highly electronegative components, RHEAs have a strong ability to regenerate passivated films in surface defects. The electronegativity values of Ti, Mo, Nb, Ta, Cr and Zr are about 1.54, 2.61, 1.60, 1.50, 1.66 and 1.33, respectively, and these can easily obtain electrons [30]. As the metal’s electronegativity increases, the electronegativity difference between the oxygen and metal decreases. Elements obey the acid–base rule, and protect the underlying metal by replicating an oxide layer to slow down the corrosion process. Cr has greater electronegativity than Ta and Zr, so it has a greater ability to regenerate passivation films. In the process of preparing a refractory high-entropy alloy coating, elements in the substrate diffuse to the coating due to the high temperature, and a small amount of Al is present in the coating. During the corrosion process, an Al_2_O_3_ passivation film is formed, but this passivation film is not very dense [31], and it is too thin due to the low content of elements. The protection it offers is not as good as that of Cr_2_O_3_, so the corrosion resistance of TC4 is relatively poor. In addition to the protective effect of the passivation film, the microstructure of the coating is also an important factor affecting the corrosion performance. As discussed in Figure 2, in +Zr, pores mainly appear at the top of the cladding zone, leading to the preferential entry of corroded Cl- into the interior of the coating along the void, which is the main reason for the low corrosion resistance of the +Zr coating. In +Ta and +Cr, pores are present at small quantities in the binding zone, which has little influence on the corrosion resistance of the coating. The three RHEAs all have BCC + HCP structures. The peak strengths of the different phases shown in Figure 1 indicate that, compared with BCC, the contents of HCP are lower, and this prevents galvanic corrosion from occurring between HCP and BCC, meaning the corrosion resistance is good. Under the action of laser beam rapid heating and supercooling, the alloying elements in the molten pool can rapidly form various compounds and the number of spontaneous nuclei can be increased, thus greatly increasing the nucleation rate, such that the microstructure becomes fine and uniform. The fine and compact structure not only reduces the content of impurities at the grain boundary; it also reduces component segregation during rapid cooling, which reduces the chance of galvanic effects occurring, thus slowing corrosion [22].

## 4. Conclusions

The three RHEAs are composed of two phases (BCC +HCP), and the main phase is BCC. The top region of the TiMoNbCr RHEA coating is mainly composed of coarse columnar dendritic β (Ti) and a rod-like second phase precipitated from a gray intercrystalline structure. The structure of the middle region of the coating is basically a basket network structure. There are a few gray columnar dendrites, basket structures and lamellar eutectic structures at the junction of the coating and the substrate. The TiMoNbTa RHEA coating is composed of columnar dendrite, a rod-like second phase and a basket network structure. The TiMoNbZr RHEA coating is mainly composed of equiaxial dendrites and columnar crystals, and has high-density defects similar to those in TC4 titanium alloy, which are composed of fine non-equiaxed dendrites and lamellar α’(Ti). Via element scanning analysis, we can infer that the coarse dendrites are in the BCC phase, the rod-like second phase is composed of HCP, and the basket structure is a full β transition structure composed of acicular α and β matrix. The corrosion resistance of the TiMoNbX (X = Cr, Ta, Zr) RHEA coating was tested in a 3.5% NaCl electrolyte solution at room temperature. It was found that, compared with the TC4 matrix, the TiMoNbX (X = Cr, Ta, Zr) RHEA coating had a higher corrosion resistance. The +Cr coating showed the strongest corrosion resistance, followed by +Zr and +Ta, and TC4 came last. These coatings form passivation films during the corrosion process. The Cr_2_O_3_ passivation films formed from +Cr coatings showed the best protective effect, while +Ta and +Zr coatings showed poor protective effects. In the +Zr coating, porosity emerged in the top region of the cladding zone, resulting in reduced corrosion resistance. Cr has greater electronegativity than Ta and Zr, so it has a greater ability to regenerate passivation films. Therefore, the conclusion is that +Cr RHEA has the best corrosion resistance.

## Figures and Tables

**Figure 1 materials-16-03860-f001:**
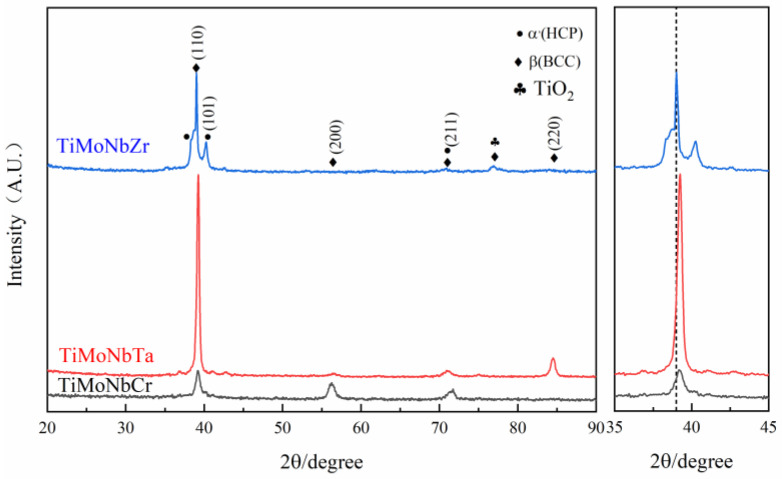
XRD pattern of TiMoNbX (X = Cr, Ta, Zr) RHEA coating.

**Figure 2 materials-16-03860-f002:**
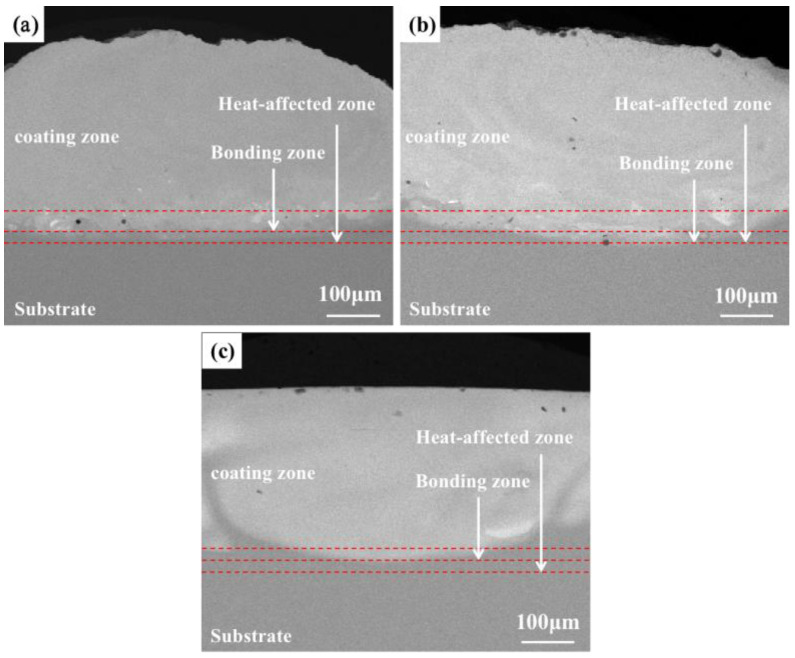
Backscatter diagram of macroscopic morphology of TiMoNbX (X = Cr, Ta, Zr) RHEA coating: (**a**) +Cr (**b**) +Ta and (**c**) +Zr.

**Figure 3 materials-16-03860-f003:**
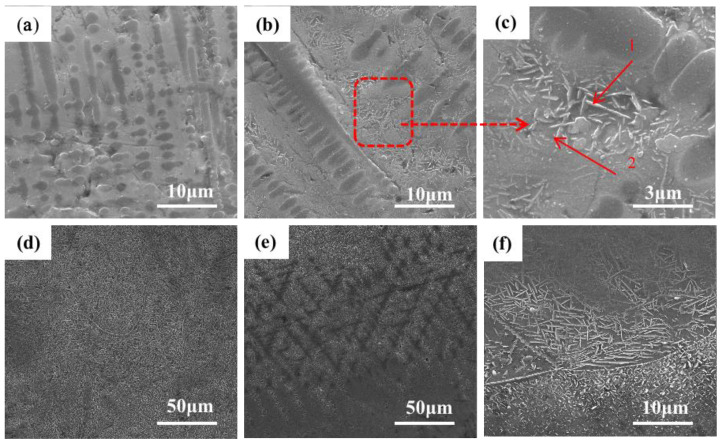
Microstructures of TiMoNbCr RHEA coating: (**a**) top; (**b**) top; (**c**) top; (**d**) middle; (**e**) bottom. (**f**) binding zone.

**Figure 4 materials-16-03860-f004:**
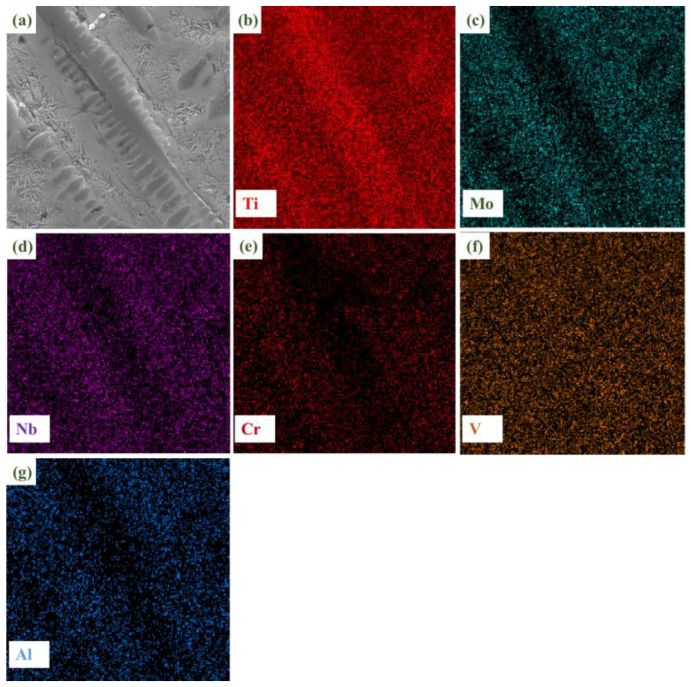
Microstructure morphology of TiMoNbCr RHEA coating and its corresponding EDS element distribution images: (**a**) microstructure morphology; (**b**) Ti; (**c**) Mo; (**d**) Nb; (**e**) Cr; (**f**) V; (**g**) Al.

**Figure 5 materials-16-03860-f005:**
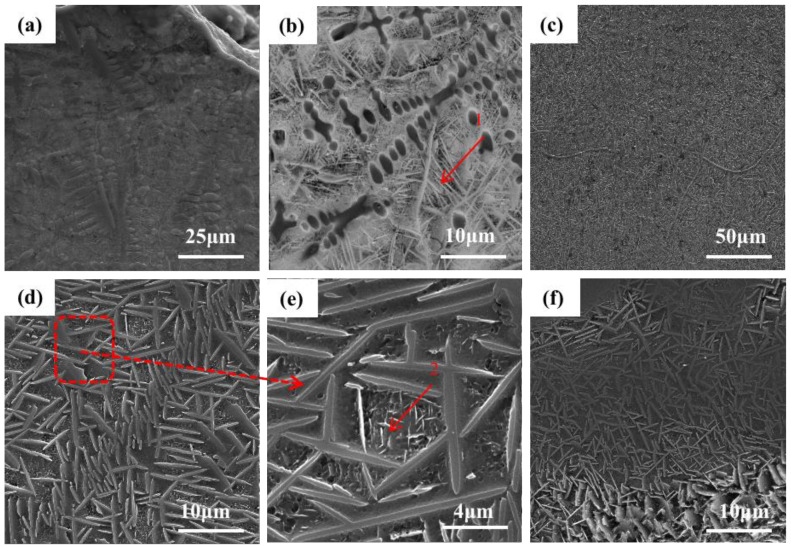
Microstructure of TiMoNbTa RHEA coating: (**a**) top; (**b**) top; (**c**) middle; (**d**) middle; (**e**) middle; (**f**) bottom.

**Figure 6 materials-16-03860-f006:**
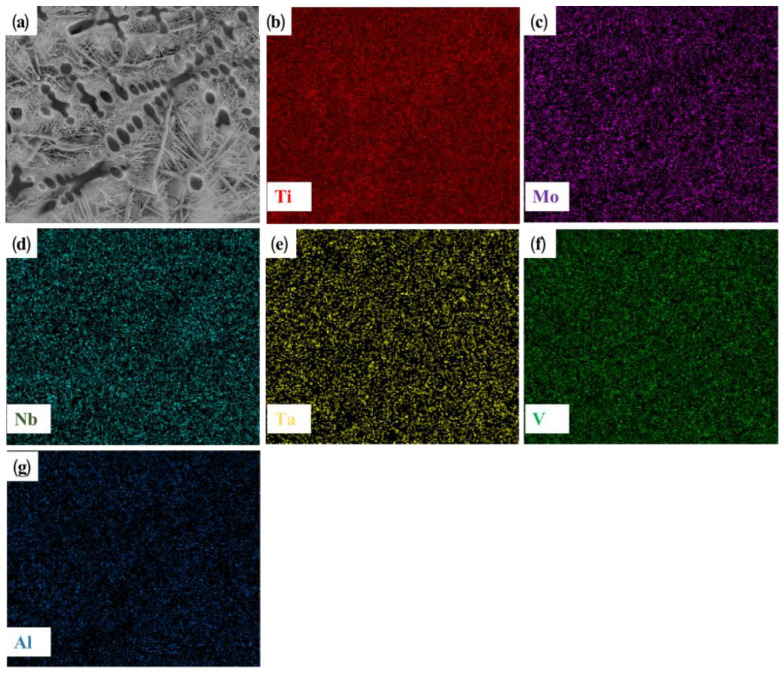
Microstructural morphology of TiMoNbTa RHEA coating and its corresponding EDS element distribution images: (**a**) microstructure morphology; (**b**) Ti; (**c**) Mo; (**d**) Nb; (**e**) Cr; (**f**) V; (**g**) Al.

**Figure 7 materials-16-03860-f007:**
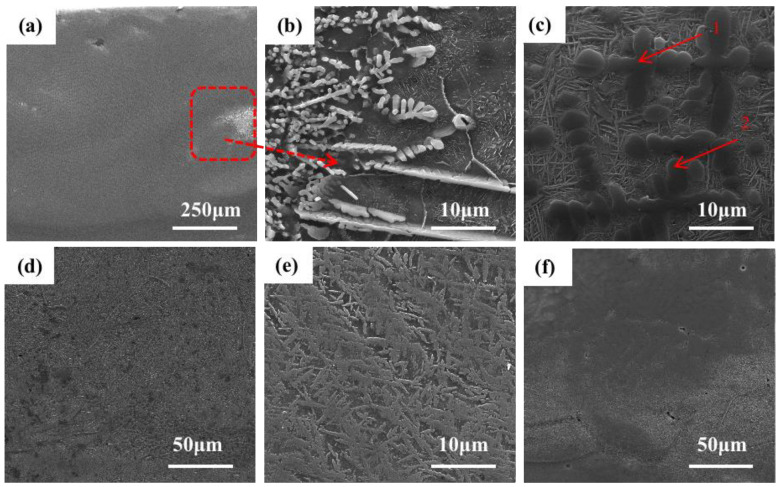
Microstructures of TiMoNbZr RHEA coating: (**a**) integral; (**b**) high-density defects; (**c**) top; (**d**) middle; (**e**) middle; (**f**) bottom.

**Figure 8 materials-16-03860-f008:**
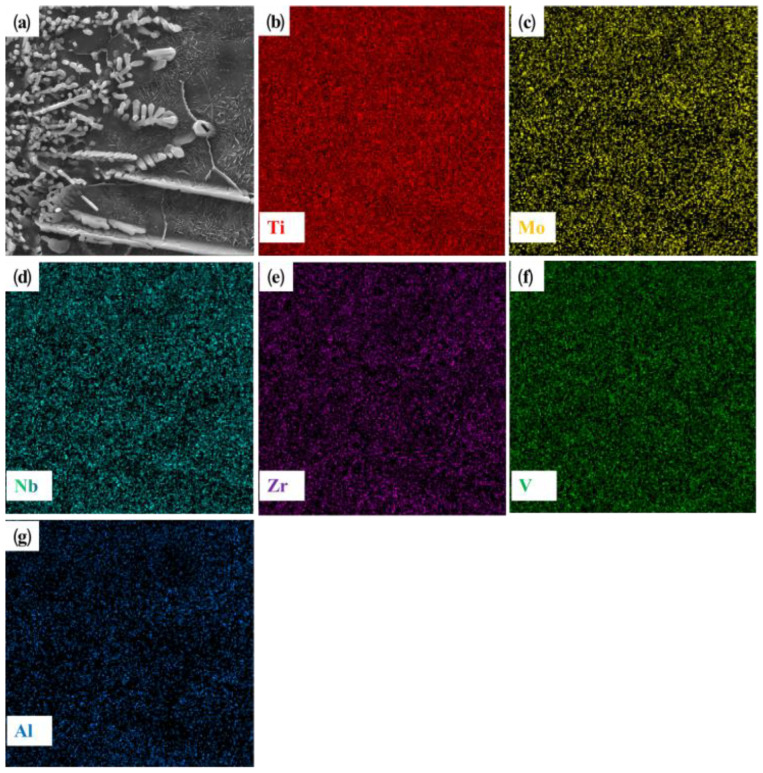
Microstructure morphology of TiMoNbZr RHEA coating and its corresponding EDS element distribution images: (**a**) microstructure morphology; (**b**) Ti; (**c**) Mo; (**d**) Nb; (**e**) Cr; (**f**) V; (**g**) Al.

**Figure 9 materials-16-03860-f009:**
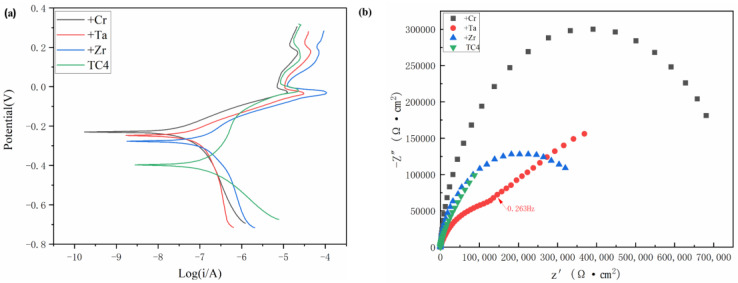
Polarization curves of TiMoNbX (X = Cr, Ta, Zr) RHEA coating and TC4 substrate in 3.5% NaCl solution and EIS diagram under OCP conditions. (**a**) Polarization curves; (**b**) EIS.

**Figure 10 materials-16-03860-f010:**
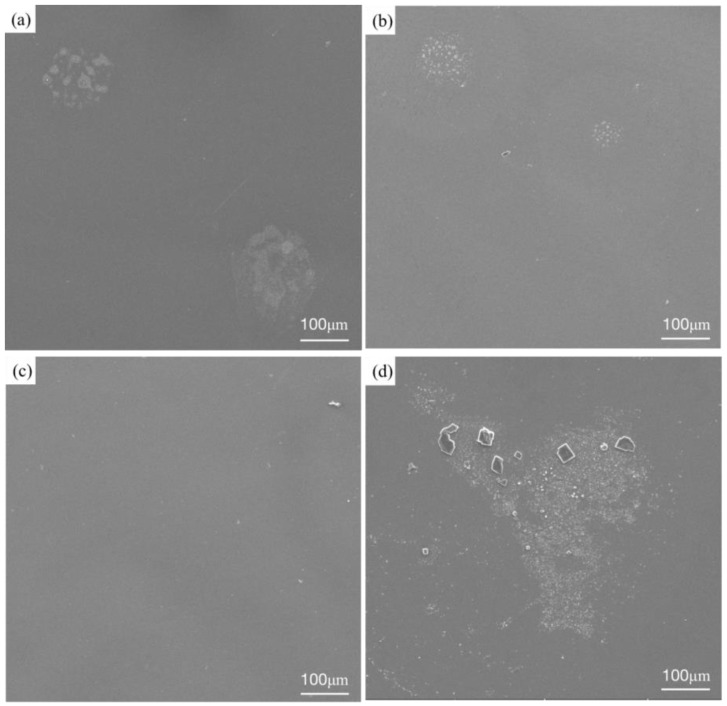
Microstructure of TiMoNbX (X = Cr, Ta, Zr) RHEA coating and TC4 matrix corroded in 3.5 mass% NaCl solution: (**a**) +Cr, (**b**) +Ta, (**c**) +Zr, (**d**) TC4.

**Figure 11 materials-16-03860-f011:**
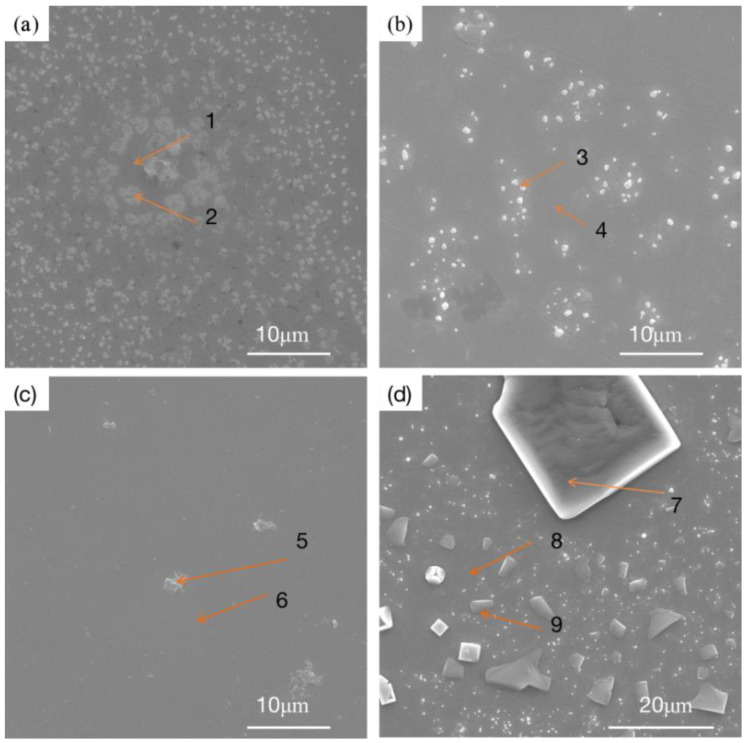
Microstructure of TiMoNbX (X = Cr, Ta, Zr) RHEA coating and TC4 matrix corroded in 3.5 mass% NaCl solution: (**a**) +Cr, (**b**) +Ta, (**c**) +Zr, (**d**) TC4.

**Table 1 materials-16-03860-t001:** Lattice constants of three RHEA coatings.

+Cr (BCC)				+Ta (BCC)			+Zr (BCC)		
No.	2θ /°	(h k l)	$a_{\left(h\;k\;l\right)}/Å$	2θ /°	(h k l)	$a_{\left(h\;k\;l\right)}/Å$	2θ /°	(h k l)	$a_{\left(h\;k\;l\right)}/Å$
1	39.2	(1 1 0)	3.250	39.2	(1 1 0)	3.250	39.0	(1 1 0)	3.263
2	56.1	(2 0 0)	3.277	56.1	(2 0 0)	3.277	71.6	(2 1 1)	3.223
3	71.6	(2 1 1)	3.223	71.6	(2 1 1)	3.223			
4	-	-	-	84.4	(2 2 0)	3.242	-	-	-
Average	-	-	3.250	-	-	3.248	-	-	3.243

**Table 2 materials-16-03860-t002:** Element compositions of marked zones of the coating in Figure 3 (at%).

Zones	Ti	Mo	Nb	Cr	V	Al
1	70.6	7.4	3.8	7.7	7.5	3.0
2	69.6	6.6	6.6	6.7	3.2	7.3

**Table 3 materials-16-03860-t003:** Element compositions of marked zones of coating in Figure 5 (at%).

Zones	Ti	Mo	Nb	Ta	V	Al
1	74.3	4.1	5.8	5.2	3.1	7.5
2	77.3	3.6	2.8	3.3	3.6	9.5

**Table 4 materials-16-03860-t004:** Element compositions of the marked zones of the coating in Figure 7c (at%).

Zones	Ti	Mo	Nb	Zr	V	Al
1	94.2	1.1	1.9	2.1	0.9	0.7
2	84.3	1.9	3.8	2.9	2.5	4.7

**Table 5 materials-16-03860-t005:** Polarization curve parameters of TiMoNbX (X = Cr, Ta, Zr) RHEA coating and TC4 matrix in 3.5% NaCl solution.

Material	E_corr_ (V)	I_corr_ (A·cm^−2^)	Epp1 (V)	Epp2 (V)
TC4	−0.397	2.797 × 10^−9^	−0.292	−0.084
TiMoNbCr	−0.230	1.722 × 10^−10^	−0.095	-------
TiMoNbTa	−0.248	1.695 × 10^−9^	−0.118	-------
TiMoNbZr	−0.277	1.788 × 10^−9^	−0.116	-------

**Table 6 materials-16-03860-t006:** Element compositions of coatings at Mark points (at%).

Point	Ti	Mo	Nb	O	V	Al	Cr	Ta	Zr	Na	Cl
1	41.1	5.6	5.3	38.2	1.9	3.7	4.3	_	_	_	_
2	53.1	7.4	6.2	19.5	2.4	5.0	6.4	_	_	_	_
3	47.3	2.4	5.2	36.6	2.1	4.0	_	2.4	_	_	_
4	1.4	6.2	5.4	19.4	3.1	7.4	_	7.1	_	_	_
5	49.1	1.8	2.0	36.2	2.1	6.8	_	_	2.0	_	_
6	71.3	2.3	1.7	10.8	3.2	8.9	_	_	1.8	_	_
7	5.2	_	_	_	0.3	0.3	_	_	_	51.7	42.5
8	83.0	_	_	2.5	5.3	9.1	_	_	_		
9	24.3	_	_	_	1.0	4.0	_	_	_	47.7	23.0

## Data Availability

Not applicable.

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
