# Peer review of "Corrosion Behavior of TiMoNbX (X = Ta, Cr, Zr) Refractory High Entropy Alloy Coating Prepared by Laser Cladding Based on TC4 Titanium Alloy"

_materials, 2023, doi:10.3390/ma16103860_

Round 1

Reviewer 1 Report

In the manuscript “Corrosion Behavior of TiMoNbX (X = Ta, Cr, Zr) refractory high entropy alloy coating prepared by laser cladding based on TC4 titanium Alloy” the authors studied the the corrosion resistance of TiMoNbX (X = Ta, Cr, Zr) quaternary refractory high entropy alloy. It was found that compared with TC4 matrix, TiMoNbX (X= Cr, Ta, Zr) refractory high entropy alloy coating had higher corrosion resistance. Among them, +Cr coating has the strongest corrosion resistance, followed by +Zr and +Ta coating, and TC4 is the last. These coatings form passivation films during the corrosion process. Cr2O3 passivation films formed by +Cr coatings have the best protective effect, while +Ta and +Zr coatings have poor protective effect. Therefore, the conclusion is that +Cr refractory high entropy alloy has the best corrosion resistance.

An important observation of the authors is the qualitative effect of porosity on the corrosion resistance of layers deposited with laser radiation, which is discussed on page 4: «By observing Fig 2 (a), (b) and (c), it can be found that there are few porosity and defects in +Ta coating, while large porosity defects appear in the cladding area of +Zr coating. In contrast, the +Cr coating has a more uniform structure. These factors have an important effect on the mechanical properties, high-temperature oxidation resistance and electrochemical corrosion properties of the coating.». However, the authors do not quantify porosity and do not mention the role of porosity in the conclusion section.

In this regard, authors are advised to:

1)    Give quantitative data on pore size and volume of porosity for +Ta coating, +Zr coating and +Cr coating.

2)    In the conclusion section, discuss the effect of porosity on the corrosion resistance of coatings in comparison with the effect of the chemical nature of coatings +Ta, +Zr and +Cr.

 The manuscript can be recommended for publication in the Journal after minor revision, eliminating the comment made.

Minor editing of English language required

Author Response

In this regard, authors are advised to:

  • Give quantitative data on pore size and volume of porosity for +Ta coating, +Zr coating and +Cr coating.

     In the process of laser cladding, the existence of pores is unavoidable. The pores of the coating prepared by us do not exist in a large area, and the performance I explore is mainly the corrosion resistance of the coating surface, and the location of pores will affect the corrosion resistance of the coating. If pores exist at the top of the coating, it will lead to poor corrosion resistance. Therefore, I did not discuss porosity in this paper, but mainly explored stomatal location. I also made a supplementary explanation on the formation of stomata in lines 330-349 of the article, please pay attention to it.

2)    In the conclusion section, discuss the effect of porosity on the corrosion resistance of coatings in comparison with the effect of the chemical nature of coatings +Ta, +Zr and +Cr.

       In line 796-800 of the paper, I made a supplementary discussion on the influence of corrosion resistance caused by stomata, please pay attention to it.

Reviewer 2 Report

The abstract, introduction, and experimentation sections are very poor and need significant improvements. However, the results are promising. But the results must be accompanied by a strong context and case building in the introduction section and a clear experimentation section. The paper cannot be accepted in its current form. The following are the comments for the authors to improve the quality of the manuscript.

Majors comments

1.       The abstract is very poor. It is not comprehensive; e.g., the authors only mentioned the TiMoNbZr refractory comparison with TC4 and did not mention anything about Cr and Ta-based refractory. The key scientific reason discovered during the study for improved corrosion behavior is not mentioned. Nothing is said about methodology.

2.       The introduction is very poor indeed. The authors have provided no context of previously reported work (state of the art) on the coating of TC4 alloys. This is against the norm of scientific writing. It gives the impression that no previous attempt has been made in this area and that the current authors are the first to do this kind of work. There are numerous papers reported in this area. The authors must add a systematic literature review after line 46 detailing what has been achieved so far in the reported study and what is the justification of their current proposed work.

3.       The experimentation part is not adequately elaborated, and there are some mistakes as well in writing, e.g., in line 60, “After mixing each powder, it was weighed according to the equal molar ratio and placed in the ball mill”. Usually, the powders are weighed before mixing.

4.       Line 51, the grind machine and paper 400 were used for grinding or polishing? Sandpapers are not used for polishing.

5.       What is the rationale for selecting TiMoNb as fixed and testing Cr, Ta, and Zr only?

6.       It is well known that powder preparation plays a crucial role in the later experimentation and the quality of the specimens produced. However, the authors have not shared any details of this process, e.g., what was the starting particle sizes and distributions, what were the mixing parameters (rpm, time, ball to powder ratio, etc.).

7.       Line 62, How the prefabrication of the layer is done? And how the 0.5 – 1 mm thickness is controlled? And most importantly, if there is such a huge variation in the thickness (0.5 mm) of the prefabricated layer, there will be an effect of it on the cooling rates during the cladding process, along with other differences.

8.       Line 64 “…then the prefabricated layer is treated at 60 ~ 64 100℃ for 3 ~ 6 hours.” How, why, and where is this process done?

9.       It is not clear why the procedure in lines 76 to 79 was done for SEM microscopy? No reason is mentioned by the authors.

10.   Fig 2 is not supporting the inferences deduced by the authors. For example, I cannot notice any heat-affected zone or bonding zone. Instead of secondary electron images, it would be helpful to provide backscattered images where different regions are automatically differentiated. Currently, the red lines are not convincingly showing the labeled regions. Fig. 2c is worse in this regard.

11.   Line 135-136, “the metallurgical bonding effect between coating and substrate is good, and no impurities” is an overclaim. These aspects cannot be viewed at this high level of zoom or scale.

12.   The authors mentioned at the end of the conclusions that Cr provided the best corrosion resistance. At the same time, the authors must comment on the high-temperature performance of the Cr compared to Ta and Zr. This is necessary as one of the motives mentioned by the authors in the introduction is that TC4 has a poor high-temperature performance, and the refractory coatings are applied to improve the high-temperature performance.

13.   After Line 130, the authors have not compared their major results/finding with the previously published work. This is regarded as discrediting the published literature.

14.   In Figures 4, 6, and 8 separately showed Ti, Mo, Nb, and other elemental maps are difficult to comprehend. Better to provide a single elemental distribution image with a color code/legend.

 Overall, the English and scientific writing of the paper is weak, and the manuscript needs professional proofing. Examples of such mistakes are but are not limited to

a.       Using the same word or phrase multiple times in a single sentence, e.g., check line 62 – 63.

b.       Substrat in Fig. 2

c.       Line 96 Figure 3.1?

d.       There is no need to mention “high entropy alloy…” again and again. The authors mentioned it five times in the Abstract only. An abbreviation can be used instead, like HEA.

Author Response

Reviewer 2

  1. The abstract is very poor. It is not comprehensive; e.g., the authors only mentioned the TiMoNbZr refractory comparison with TC4 and did not mention anything about Cr and Ta-based refractory. The key scientific reason discovered during the study for improved corrosion behavior is not mentioned. Nothing is said about methodology.

    I have supplemented and revised the content of the article, and rewrote the content of the abstract. Please pay attention to it.

  1. The introduction is very poor indeed. The authors have provided no context of previously reported work (state of the art) on the coating of TC4 alloys. This is against the norm of scientific writing. It gives the impression that no previous attempt has been made in this area and that the current authors are the first to do this kind of work. There are numerous papers reported in this area. The authors must add a systematic literature review after line 46 detailing what has been achieved so far in the reported study and what is the justification of their current proposed work.

    The corrosion resistance of refractory high entropy alloy is supplemented in line 124-164 of the article, please pay attention to it.

  1. The experimentation part is not adequately elaborated, and there are some mistakes as well in writing, e.g., in line 60, “After mixing each powder, it was weighed according to the equal molar ratio and placed in the ball mill”. Usually, the powders are weighed before mixing.

    I have supplemented and modified the experimental explanation, please pay attention to lines 172-211 of the paper.

  1. Line 51, the grind machine and paper 400 were used for grinding or polishing? Sandpapers are not used for polishing.

    It has been revised. Please look at line 173 of the paper.

  1. What is the rationale for selecting TiMoNb as fixed and testing Cr, Ta, and Zr only?

    I added the reasons for choosing this system in line 165-169 of the paper. In the supplementary research background (line 124-163 of the paper), Cr, Ta and Zr all play a role in improving the corrosion resistance of the coating, so I chose to add these three elements, and then explored which coating has the best corrosion resistance.

  1. It is well known that powder preparation plays a crucial role in the later experimentation and the quality of the specimens produced. However, the authors have not shared any details of this process, e.g., what was the starting particle sizes and distributions, what were the mixing parameters (rpm, time, ball to powder ratio, etc.).

    I have added an explanation. Please refer to lines 188-197 of the paper.

  1. Line 62, How the prefabrication of the layer is done? And how the 0.5 – 1 mm thickness is controlled? And most importantly, if there is such a huge variation in the thickness (0.5 mm) of the prefabricated layer, there will be an effect of it on the cooling rates during the cladding process, along with other differences.

    The thickness of the preset layer has been modified. The thickness of the preset layer is an approximate value and there may be some errors. I will make a supplementary explanation on the experimental procedure.

  1. Line 64 “…then the prefabricated layer is treated at 60 ~ 64 100℃for 3 ~ 6 hours.” How, why, and where is this process done?

    I have added an explanation. Please refer to lines 188-197 of the paper.

  1. It is not clear why the procedure in lines 76 to 79 was done for SEM microscopy? No reason is mentioned by the authors.

    I have added an explanation. Please refer to lines 207-208 of the paper.

  1. Fig 2 is not supporting the inferences deduced by the authors. For example, I cannot notice any heat-affected zone or bonding zone. Instead of secondary electron images, it would be helpful to provide backscattered images where different regions are automatically differentiated. Currently, the red lines are not convincingly showing the labeled regions. Fig. 2c is worse in this regard.

I replaced Fig 2 with a backscatter diagram that you can check out

  1. Line 135-136, “the metallurgical bonding effect between coating and substrate is good, and no impurities” is an overclaim. These aspects cannot be viewed at this high level of zoom or scale.

    As I recast this passage, please refer to lines 336 to 337 of the paper.

  1. The authors mentioned at the end of the conclusions that Cr provided the best corrosion resistance. At the same time, the authors must comment on the high-temperature performance of the Cr compared to Ta and Zr. This is necessary as one of the motives mentioned by the authors in the introduction is that TC4 has a poor high-temperature performance, and the refractory coatings are applied to improve the high-temperature performance.

    I have revised the introduction part. This article is mainly about corrosion resistance.

  1. After Line 130, the authors have not compared their major results/finding with the previously published work. This is regarded as discrediting the published literature.

    In the corrosion resistance research section of the article, I have added relevant references. Please refer to lines 467, 588, 781, 800 and 816 of the article.

  1. In Figures 4, 6, and 8 separately showed Ti, Mo, Nb, and other elemental maps are difficult to comprehend. Better to provide a single elemental distribution image with a color code/legend.

    I modified Figs 4, 6, and 8 to match the color of the element character to the color that represents the element in the picture.

Reviewer 3 Report

1. Write better the paragraph 33.

2. Use the same criteria to name the figures (Fig. o Figure).

3. If is possible that the images of Figure 2 are changed with respecto to the test?.

4. In general the quality of images is very poor. I cant see anything in EDX images.

5. There are several repeated paragraphs.

Author Response

  1. Write better the paragraph 33.

    Paragraph 33 has been rewritten. Please refer to lines 32-39 of the paper.

  1. Use the same criteria to name the figures (Fig. o Figure).

    All images have been renamed to the same standard (Fig.).

  1. If is possible that the images of Figure 2 are changed with respecto to the test?.

    I replaced Fig 2 with a backscatter diagram that you can check out.

  1. In general the quality of images is very poor. I cant see anything in EDX images.

    I modified Figs 4, 6, and 8 to match the color of the element character to the color that represents the element in the picture.

  1. There are several repeated paragraphs.

    After line 633, there are repeated paragraphs. I have deleted them and added amendments to the content. Please pay attention to lines 772-796 of the article.

Reviewer 4 Report

This paper has presented the corrosion Behavior of TiMoNbX (X = Ta, Cr, Zr) refractory high entropy alloy coating prepared by laser cladding based on TC4 3 titanium Alloy.

1) The authors should provide a more comprehensive discussion about the various aspects of TC4 and high entropy alloys. Below are a few references to help with this process:

 - Advanced Materials Research 2012 (Vol. 566, pp. 466-469). Trans Tech Publications Ltd.

- Journal of Alloys and Compounds, (2023) 170091

2) Based on the XRD results presented in Fig. 1, please report the volume fractions of each phase.

3) Please provide more explanation about the formation of α'(Ti) during rapid cooling.

4) It is recommended to calculate the lattice distortion of different coatings based on the misfit parameter and the difference in electronegativity

5) What are the mechanisms for the formation of pores in applied coatings?

6) As shown in Fig. 4, what is the effect of vanadium solution on RHEA coatings?

Moderate editing of English language

Author Response

1) The authors should provide a more comprehensive discussion about the various aspects of TC4 and high entropy alloys. Below are a few references to help with this process:

 - Advanced Materials Research 2012 (Vol. 566, pp. 466-469). Trans Tech Publications Ltd.

- Journal of Alloys and Compounds, (2023) 170091

The corrosion resistance of refractory high entropy alloy is supplemented in line 124-164 of the article, please pay attention to it.

  • Based on the XRD results presented in Fig. 1, please report the volume fractions of each phase.

    I have read the articles about corrosion resistance of RHEA carefully, and the description of phase volume fraction has little relation to corrosion resistance, and only Ti phase can be seen in XRD, which is due to the dilution of elements in the matrix and the high Ti content of the alloy itself. In terms of corrosion resistance, it is mainly the influence of passivated elements. Therefore, I think it is unnecessary to report the phase volume fraction of Ti here. Thank you very much for your suggestion.

  • Please provide more explanation about the formation of α'(Ti) during rapid cooling.

    The formation of α' (Ti) during rapid cooling has been explained in lines 278-284.

  • It is recommended to calculate the lattice distortion of different coatings based on the misfit parameter and the difference in electronegativity。

    According to your suggestion, I made a supplementary explanation of lattice parameters in the 316-326 lines of the article. Detailed data are listed in Table 1, please pay attention to it.

  • What are the mechanisms for the formation of pores in applied coatings?

    The cause of stomatal formation is in line 399-346 of the article. According to your suggestion, I make a supplementary explanation in line 351-374 of the article, please pay attention to it.

  • As shown in Fig. 4, what is the effect of vanadium solution on RHEA coatings?

    The addition of Cr,Ta and Zr has little effect on the microstructure of the coating, and their microstructure is roughly the same, except that Zr addition has a high density defect similar to that of TC4 titanium alloy, which is described in article 538-541. It has been added in the background introduction that TiMoNb alloy has a single-phase BCC single-phase structure. These three four-element RHEAs become biphase structure (BCC+HCP). The most important thing is that these three elements are to improve the corrosion resistance of the coating. The influence of these three elements on the corrosion resistance of the coating is studied later.

Reviewer 5 Report

Paper is devoted to the formation of the protective coatings on the TC4 titanium alloy.  Manuscript needs the improvement before publication. 1) Section 2.1. Materials preparation, line 63, 64, how was the prefabricated layer formed? 2) Section 2.2. Electrochemical corrosion test, lines 91, 92, the electrolyte composition should be describes in this section, and the 3.5 % NaCl solution is not saturated solution. 3) Line 96, citation to Fig. 3.1 is wrong. 4) Correct the sentence within lines 311,312. 5) Delete the duplicate text on the pages 12, 13, 14, 15.

Author Response

  • Section 2.1. Materials preparation, line 63, 64, how was the prefabricated layer formed?

    The formation of the preset layer is further explained in line 188-198 of the paper, please pay attention to it.

  • Section 2.2. Electrochemical corrosion test, lines 91, 92, the electrolyte composition should be describes in this section, and the 3.5 % NaCl solution is not saturated solution

    It has been revised. Please refer to lines 224-226 of the paper.

  • Line 96, citation to Fig. 3.1 is wrong.

    It has been revised. Please refer to line 264 of the paper.

  • Correct the sentence within lines 311,312.

    This has been corrected. Please refer to line 591 of the paper.

  • Delete the duplicate text on the pages 12, 13, 14, 15.

    The repeated paragraphs have been corrected. Please refer to lines 616-635 of the paper.

Round 2

Reviewer 2 Report

The paper has been improved than before. However, there are still some flaws that must be corrected.

1.       Line 69 to Line 109, the authors have added some literature review but it is not well written. E.g., it is not clear what is the base material for most of the studies reviewed. The authors must review literature focused more on coatings done on the TC4 substrate and clearly highlight in the last paragraph of the intro what is the gap that has not been addressed in the previous studies.

2.       The authors have used many unknown (undefined terminologies in the Line 69 to Line 109, this again illustrates poor scientific writing.

3.       Line 110 – 112 is looking out of context, I am unable to understand what the authors are trying to mention here. This need to be elaborated in detail in a clearer way.

4.       The authors have not paid much attention while answering the comments and line numbers. They are referring me to wrong line numbers, e.g. see below for comment 3

“The experimentation part is not adequately elaborated, and there are some mistakes as well in writing, e.g., in line 60, “After mixing each powder, it was weighed according to the equal molar ratio and placed in the ball mill”. Usually, the powders are weighed before mixing.

I have supplemented and modified the experimental explanation, please pay attention to lines 172-211 of the paper.”

5.       The same is true for comments 4, 5, …. This wastes my time, and I can’t continue to review the revision like this. Please provide your answer with the correct line numbers for all of your answers.

6. The new figure 2 does not show any details of the previously claimed bonding zone, HAZ, and coating. Add proper details and annotations in the correct figures. Also, why the coatings are appearing as a convex/concave form instead of a layer?

English and scientific writing still need to be improved.

Author Response

  1. The abstract is very poor. It is not comprehensive; e.g., the authors only mentioned the TiMoNbZr refractory comparison with TC4 and did not mention anything about Cr and Ta-based refractory. The key scientific reason discovered during the study for improved corrosion behavior is not mentioned. Nothing is said about methodology.

I have supplemented and revised the content of the article, and rewrote the content of the abstract. Please pay attention to it.

  1. The experimentation part is not adequately elaborated, and there are some mistakes as well in writing, e.g., in line 60, “After mixing each powder, it was weighed according to the equal molar ratio and placed in the ball mill”. Usually, the powders are weighed before mixing.

I have supplemented and modified the experimental explanation, please pay attention to lines 87 of the paper.

  1. Line 51, the grind machine and paper 400 were used for grinding or polishing? Sandpapers are not used for polishing.

It has been revised. Please look at line77- 78 of the paper.

  1. What is the rationale for selecting TiMoNb as fixed and testing Cr, Ta, and Zr only?

I added the reasons for choosing this system in line 66 of the paper. Cr, Ta and Zr all play a role in improving the corrosion resistance of the coating, so I chose to add these three elements, and then explored which coating has the best corrosion resistance.

  1. It is well known that powder preparation plays a crucial role in the later experimentation and the quality of the specimens produced. However, the authors have not shared any details of this process, e.g., what was the starting particle sizes and distributions, what were the mixing parameters (rpm, time, ball to powder ratio, etc.).

I have added an explanation. Please refer to lines 86-92 of the paper

  1. Line 62, How the prefabrication of the layer is done? And how the 0.5 – 1 mm thickness is controlled? And most importantly, if there is such a huge variation in the thickness (0.5 mm) of the prefabricated layer, there will be an effect of it on the cooling rates during the cladding process, along with other differences.

The thickness of the preset layer has been modified. The thickness of the preset layer is an approximate value and there may be some errors. I will make a supplementary explanation on the experimental procedure. Please look at line 94 of the paper

  1. Line 64 “…then the prefabricated layer is treated at 60 ~ 64 100℃for 3 ~ 6 hours.” How, why, and where is this process done?

I have added an explanation. Please refer to lines 95-96 of the paper.

  1. It is not clear why the procedure in lines 76 to 79 was done for SEM microscopy? No reason is mentioned by the authors.

I have added an explanation. Please refer to lines 105-107 of the paper

  1. Fig 2 is not supporting the inferences deduced by the authors. For example, I cannot notice any heat-affected zone or bonding zone. Instead of secondary electron images, it would be helpful to provide backscattered images where different regions are automatically differentiated. Currently, the red lines are not convincingly showing the labeled regions. Fig. 2c is worse in this regard.

I replaced Fig 2 with a backscatter diagram that you can check out.

  1. Line 135-136, “the metallurgical bonding effect between coating and substrate is good, and no impurities” is an overclaim. These aspects cannot be viewed at this high level of zoom or scale.

As I recast this passage, please refer to lines 176 to 178 of the paper

  1. The authors mentioned at the end of the conclusions that Cr provided the best corrosion resistance. At the same time, the authors must comment on the high-temperature performance of the Cr compared to Ta and Zr. This is necessary as one of the motives mentioned by the authors in the introduction is that TC4 has a poor high-temperature performance, and the refractory coatings are applied to improve the high-temperature performance.

I have revised the introduction part. This article is mainly about corrosion resistance.

  1. After Line 130, the authors have not compared their major results/finding with the previously published work. This is regarded as discrediting the published literature.

In the corrosion resistance research section of the article, I have added relevant references. Please refer to lines 171, 257, 343, 392 and 400 of the article.

  1. In Figs 4, 6, and 8 separately showed Ti, Mo, Nb, and other elemental maps are difficult to comprehend. Better to provide a single elemental distribution image with a color code/legend.

I modified Figs 4, 6, and 8 to match the color of the element character to the color that represents the element in the picture.

  1. Line 69 to Line 109, the authors have added some literature review but it is not well written. E.g., it is not clear what is the base material for most of the studies reviewed. The authors must review literature focused more on coatings done on the TC4 substrate and clearly highlight in the last paragraph of the intro what is the gap that has not been addressed in the previous studies.

There are indeed studies on the corrosion resistance of RHEA on TC4 matrix, but the number is not much, and there are fewer related to the elements I study. Secondly, I mainly study the corrosion resistance of refractory high entropy alloy coating, with the purpose of improving the corrosion resistance of TC4, so most of the articles I refer to are about the corrosion resistance of RHEA. So I can find more articles related to my alloying elements. However, I have added relevant references such as [19].

  1. The new figure 2 does not show any details of the previously claimed bonding zone, HAZ, and coating. Add proper details and annotations in the correct figures. Also, why the coatings are appearing as a convex/concave form instead of a layer?

I have modified it according to your meaning.

  1. Line 110 – 112 is looking out of context, I am unable to understand what the authors are trying to mention here. This need to be elaborated in detail in a clearer way.

I have modified it according to your meaning, please pay attention to the lines 46-73.

Reviewer 3 Report

The author has greatly improved his manuscript by attending to the indications made by the reviewer

Author Response

I have modified the format and some syntax errors according to your requirements.

Reviewer 4 Report

The revision is satisfactory.

Author Response

I have made some changes to the format and introduction of the paper, which you may have a look at.

Round 3

Reviewer 2 Report

The manuscript has been significantly improved and can be published in its present form.